# Fuzzy Logic and Decision Making Applied to Customer Service Optimization

**Gabriel Marín Díaz** * and **Ramón Alberto Carrasco González**

Faculty of Statistics, Complutense University Puerta de Hierro, 28040 Madrid, Spain
* Correspondence: gmarin03@ucm.es

**Abstract:** In the literature, the Information Technology Infrastructure Library (ITIL) methodology recommends determining the priority of incident resolution based on the impact and urgency of interactions. The RFID model, based on the parameters of Recency, Frequency, Importance and Duration in the resolution of incidents, provides an individual assessment and a clustering of customers based on these factors. We can improve the traditional concept of waiting queues for customer service management by using a procedure that adds to the evaluation provided by RFID such additional factors as Impact, Urgency and Emotional character of each interaction. If we also include aspects such as Waiting Time and Contact Center Workload, we have a procedure that allows prioritizing interactions between the customer and the Contact Center dynamically and in real time. In this paper we propose to apply a model of unification of heterogeneous information in 2-tuple linguistic evaluations, to obtain a global evaluation of each interaction by applying the Analytic Hierarchy Process (AHP), and in this way be able to have a dynamic process of prioritization of interactions.

**Keywords:** fuzzy logic; machine learning; AHP; RFID; making-decisions; contact center; customer value

**MSC:** 03E72; 03B52; 68T27; 68T20

## 1. Introduction

The advance of digitalization motivates companies to be in a continuous process of digital transformation, and customer service is one of the basic pillars in this process. A piece of information that almost all companies have in their systems corresponds to the purchase history of their customers, which is why applying the methodology based on Recency, Frequency and Monetary transactions for a considered period (RFM) [1], is relatively easy to execute and implement. This model allows us to know the Customer Lifetime Value (CLV) [2], and thus build customer profiles according to the brand's valuation of the customer, and even determine which marketing actions may be more appropriate according to this valuation.

If we focus on the Contact Center (Customer Service), companies have the data for the interactions that the customer makes through it. They also have the customer's assessment of the brand through a metric such as the Net Promote Score (NPS) [3], or through the Customer Satisfaction Score (CSAT) [4] and a measure of customer effort, Customer Effort Score (CES) [5]; the basis of this approach focuses on enhancing the value of customer interaction in a simple way, prioritizing solutions over speed. The question is how to optimize and integrate all this information generated in the sales and post-sales phase to increase customer engagement and therefore sales volume.

Social networks determine that any information that is shared by the customer in a public way and, therefore, any positive or negative opinion, spreads exponentially through different channels. That is why the value of the customer should not be limited simply to the purchase history—the impact that the customer can have on the brand in social networks

can be decisive. The study conducted by Kumar [6] goes beyond the transactional aspect and integrates communication channels to suggest the adoption of a multichannel and multimedia strategic framework, which focuses on customer preferences and the creation of value for them, and introduces the concept of Customer Engagement Value (CEV), which is composed of four variables: the first, the mentioned CLV, based on customer buying behavior; the Customer Referral Value (CRV) based on the referral value of customer opinions; the Customer Influencer Value (CIV) based on the value of customer influence towards other customers and finally the Customer Knowledge Value (CKV) based on the value added to the company by customer feedback. In this paper we extend the CEV model with the addition of the RFID model, based on the Recency, Frequency, Importance and Duration of customer interactions with the Contact Center, to obtain the metric called the Customer Service Value (CSV) in a linguistic domain of representation, 2-tuple [7].

Premier manufacturers such as Salesforce [8], have tools integrated in the functionality of Customer Relationship Management (CRM) that allow to develop with guarantees all the operational processes related to customer management, including marketing, ecommerce, sales and service. The main features of CRM focused on the Contact Center are aimed at managing better service demand, adequate staff sizing, prioritization of interactions, and the development of a multichannel service. The main objective of all these actions is to reduce the average time of operation (TMO) in each interaction and consequently improve customer satisfaction levels.

The measurement of service quality, understood as the difference between the value expected by the customer and the value delivered by the brand, is closely related to the Service Level Agreement (SLA) [9], and consequently to the priority and speed with which incidents are attended. In addition, a fundamental aspect in this prioritization is the degree of personalization in communication. Information Technology Infrastructure Library (ITIL) methodology [10,11] recommends prioritizing the resolution of incidents based on impact and urgency.

These ideas led us to investigate and develop further in this research what had been done previously in relation to this purpose, and how the relationship between customer and brand could be improved by focusing on the Contact Center. In an exhaustive review of the literature, no research was found that addressed this problem considering the workload of the Contact Center, deriving the interactions for the agent resulting in less work, or for an automated communication channel, beyond the classic queue management. On the other hand, technological tools are oriented in the same direction.

This gave rise to a research program related to the processes of prioritization and personalization of interactions, based on criteria such as Customer Service Value (V), Impact (I), Urgency (U) and the Emotional (E) nature of each interaction (VIUE), thus expanding the concept of classic queue management.

The value of the customer is obtained directly from the Contact Center's evaluation of the customer through the history of their interactions, the RFID model [7], Impact and Urgency (linguistic variables) collected directly from the customer management system of any company (CRM). Additionally, the assessment of the emotional character (linguistic variable) of the interaction is obtained from the realization of a sentiment analysis in the interaction process, which affect the processes and the outcome of the service offered by the agent attending the incident [12].

On the other hand, customer scoring is a live process. We are continuously receiving interactions in real time and, therefore, the Contact Center activity queue is subject to changes. It could be the case that, if we only classify the interactions by the criteria defined above, we could leave customers with a low priority rating, and they remain unattended. To avoid this, a second classification is made, which depends on the SLAs defined for the customer or customer segment, i.e., the maximum waiting time (T) and Contact Center workload (C) [9], both numerical variables, thus obtaining an additional phase of contextual adjustment of the model.

With respect to the proposed methodology, it has been considered that most companies that have grown under a CRM management application dispose of the customer's basic data, their purchase history through the RFM model, and their interaction history with the Contact Center through the RFID model. Thus, in the methodology proposed in this research, the improvements developed in the model RFID take on special relevance [7], introducing the 2-tuple model to solve the problems of lack of accuracy of heterogeneous information processing. This will help us in the process of representation of linguistic information [13], and thus improve the processing of heterogeneous information [14], and unify different types of information, numerical and linguistic, so that the scoring of each client can be obtained. Furthermore, the multi-criteria decision making model (AHP), proposed by Thomas L. Saaty in 1980 [15], will help us to determine the weight of each criterion in the proposed model (VIUE) and, therefore, to obtain the final score that will determine the prioritization of each interaction.

As the main contributions of the paper, we can consider the following:

- The availability of a methodology based on fuzzy logic and multi-criteria decision-making that allows real-time prioritization of tickets according to variables such as customer value, impact, urgency, and the emotional nature of each interaction.
- Contextualize the model in different usage scenarios, considering additional variables such as waiting time and contact center workload. This allows a reordering process to be carried out according to these variables, and thus comply with the established SLAs.
- Develop a model that allows the weighting of variables in real time, and therefore, a dynamic adaptation to the particularities of the business.
- Extend a working methodology based on multi-criteria decision-making in the Customer Service area.

The rest of the paper is organized as follows. In Section 2, we will briefly review the related literature; in Section 3, we will detail the theoretical research framework; in Section 4, we will develop the elements that make up the VIUE model; in Section 5, we will detail the proposed model applied to a software manufacturing company; and finally in Sections 6–8, we will present the conclusions and future work.

## 2. Literature Review

In the field of marketing science, measuring Customer Satisfaction is a critical metric and countless studies refer to the need to properly manage the relationship between the customer and the brand. The study [16], covers the five key dimensions of perceived service quality measurement: reliability, assurance, tangibility, empathy and responsiveness. An extension of the previous study [17], highlights the importance of the following dimensions in the process of developing strong customer-brand relationships: reliability, empathy, customer knowledge, customer orientation, waiting time, ease of use and accessibility.

Utilizing the quality and customer satisfaction model as a framework, we can emphasize the following metric as one of the most widely used, the NPS proposed by F. Reichheld [3], which uses the value of customer referral as a measure of loyalty. Among studies similar to this one is the CES score [5], which is based on the idea that customer interactions should be simple, prioritizing the solution over any other factor. In the study [18], the authors noted that while these metrics have some intuitive power, they lack sound theoretical development, focus on a specific CES domain or focus on ad hoc NPS operations, and stressed that metrics that assess customer behavior from a 360° predict customer behavior better than a single metric.

In recent years, techniques for data mining have been used to perform customer segmentation processes, for example, k-means, logistic regression, neural networks, etc. However, the trend in the marketing environment is to use RFM models in conjunction with other models, mainly because of its easy interpretability and the possibility of making explainable decisions [19]. However, we did not find an article that rated customer ratings from the perspective of their relationship with the Contact Center. In the VIUE model, the customer rating is based on the RFID model that does take this metric into account [7], in

addition to the ITIL methodology for measuring the impact and urgency of: the interaction [10,11,20]; the interaction sentiment analysis [21–23]; and the workload and response time established by the SLA [9].

Figure 1 lists the publications and citations related to the following search variables: TS = (CUSTOMER SERVICE) AND TS = (PRIORITY) AND TS = (CONTACT CENTER). The objective was to discover the scientific publications related to the management of priorities in customer interactions with the Contact Center. As can be seen, the number of publications since 2017 was nine, Table 1. In Figure 2, the total number of publications can be seen without limitation of dates, 33; and Table 2 shows these publications classified by category.

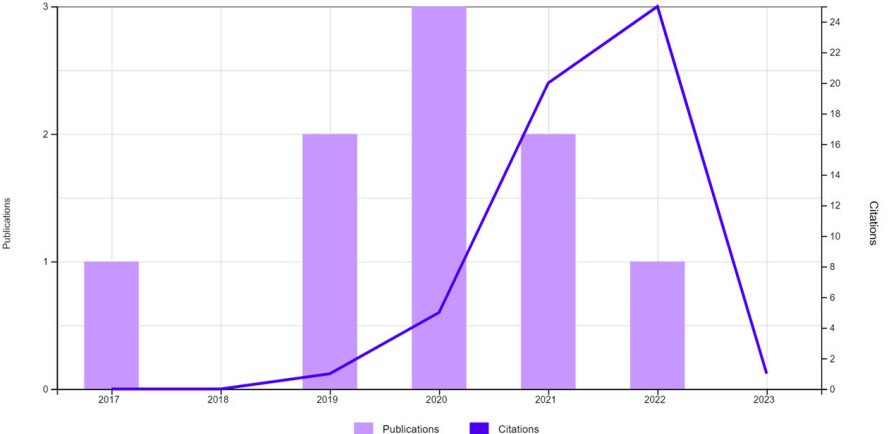

**Figure 1.** Studies Priority Interactions Contact Center, 2017–2023.

**Table 1.** Studies Priority Interactions Contact Center, 2017–2023.

| Category | Title |
|---|---|
| Queue and Routing | The Economics of Line-Sitting [24] |
| | A Model of Queue Scalping [25] |
| | Queuing System with Two Types of Customers and Dynamic Change of a Priority [26] |
| Priority | Priority Service Pricing with Heterogeneous Customers: Impact of Delay Cost Distribution [27] |
| | Personalized Priority Policies in Call Centers Using Past Customer Interaction Information [28] |
| Clustering | Multi-attribute intelligent queueing method for onboard call centers [29] |
| | General Practice and the Community: Research on health service, quality improvements and training [30] |
| Quality Service | The Dispositional Attribution of Customer Satisfaction through the Juxtaposition of QFD and Servqual in Service Industry Design [31] |
| | How Amazon went from an uncertain online bookstore to the leader in e-commerce [32] |

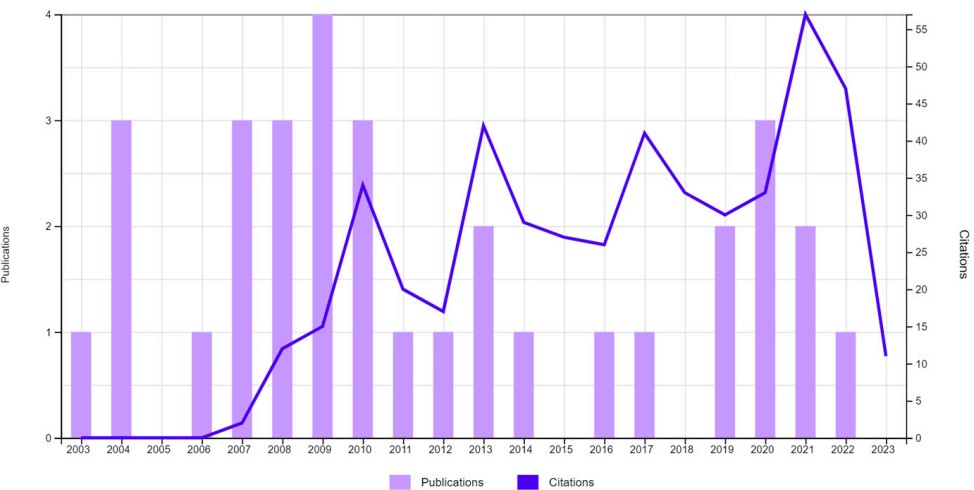

**Figure 2.** Studies related to the Priority of Interactions in Contact Centers.

**Table 2.** Studies related to the Priority of Interactions in Contact Centers.

| Category | Publications | % |
|---|---|---|
| Queue and Routing | 15 | 45.45% |
| Service Level | 12 | 36.36% |
| Training | 3 | 9.09% |
| Personalization | 3 | 9.09% |

For Fuzzy Logic and decision-making methods applied to Contact Centers, TS = (CONTACT CENTER) AND TS = (FUZZY LOGIC), we found two papers, the first from 2006 [33], which refers to the development of a fuzzy expert system methodology to categorize customers and customer service agents. The next study [34], from 2009, refers to interpretable classification systems based on fuzzy rules and applied to data extracted from a customer service center.

The publications related to the study objective of this article were scarce, mainly based on routing to the operator with the lowest workload and on the management of messaging queues. The proposed model presents a notable novelty in the way customer incidents are prioritized, escalated, and handled.

## 3. Methodology

This section addresses the theoretical framework of the research. For this purpose, the following models will be used: the fuzzy 2-tuple linguistic model (LD2T), the analytic hierarchy process (AHP), and the treatment of heterogeneous information in the decision-making process.

In many cases the information necessary for decision making is not represented in the same domain of expression. If this is the case, some criteria involved in the decision-making process may not be quantifiable in numerical values, thus presenting imprecision and therefore subjectivity. In these cases, it will be necessary to use a model that allows to obtain intermediate and global valuations that are interpretable under the same domain of expression. This is the reason why we will use a linguistic domain [35] in order to unify the information processed in the VIUE model.

### 3.1. 2-Tuple Model (LD2T)

The 2-tuple model based on the unification of information into linguistic values was proposed by F. Herrera and L. Martinez [13]. The purpose of this model is to improve the information loss problem in the computation process with linguistic labels. The following briefly introduces the 2-tuple linguistic representation model and its computation system. The model is based on a pair of represented values $(s_i, \alpha_i)$, where $s_i \in S$ and $\alpha_i \in [-0.5, 0.5]$.

The membership function chosen corresponds to a triangular function, a representation of such a domain in $S5$ is shown in Figure 3.

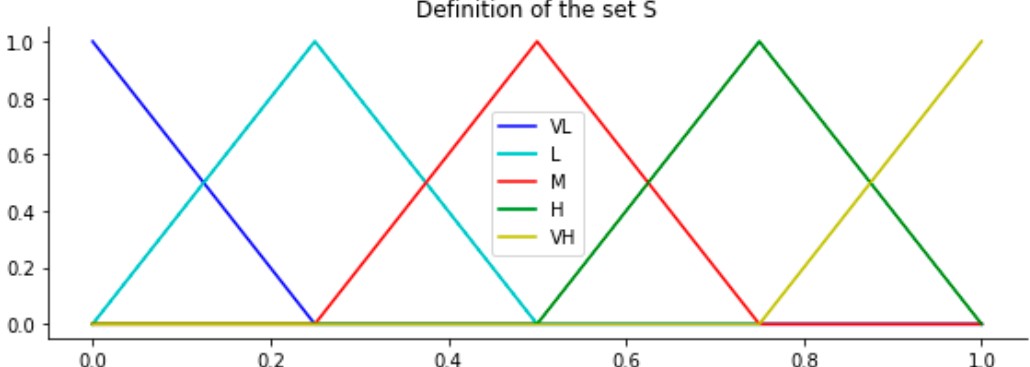

**Figure 3.** Definition of the set $S$.

**Definition 1.** *Let* $S = \{s_0, \ldots, s_p\}$ *a set of linguistic terms with odd cardinality, and* $\beta \in [0, p]$ *a value in the granularity range of* $S$. *Assuming further that the labels are represented by triangular functions, the symbolic translation of a linguistic item* $s_i$ *is a number in the interval* $[-0.5, 0.5)$, *indicates the difference between a set of information, represented by the value of* $\beta \in [0, p]$ *obtained in a symbolic operation and the nearest integer value,* $i \in \{0, \ldots, p\}$.

**Definition 2.** *Given a set of linguistic terms* $S = \{s_0, \ldots, s_p\}$, $\langle S \rangle = S \times [-0.5, 0.5)$, *and a value* $\beta \in [0, p]$ *that represents the outcome of a symbolic operation, the linguistic 2-tuple equivalent to* $\beta$ *can be determined using the following function:*

$$\Delta_S : [0, p] \to \langle S \rangle$$
$$\Delta_S(\beta) = (s_i, \alpha_i), \quad \begin{cases} i = round(\beta) \\ \alpha = \beta - i, \alpha \in [-0.5, 0.5), \end{cases} \tag{1}$$

*where* $round(\cdot)$ *is the usual round operation,* $s_i$ *is the label with index closest to* $\beta$ *and* $\alpha$ *is the value of the symbolic translation. Thus, a value in the interval* $[0, p]$ *is identified by a 2-tuple in the set* $\langle S \rangle$.

**Definition 3.** *Let* $S = \{s_0, \ldots, s_p\}$ *a set of linguistic terms and* $(s_i, \alpha_i) \in \langle S \rangle = S \times [-0.5, 0.5)$. *The numerical value in the granularity range* $[0, p]$ *which represents the linguistic value 2-tuple* $(s_i, \alpha_i)$ *is obtained using the function:*

$$\Delta_S^{-1} : \langle S \rangle \to [0, p]$$
$$\Delta_S^{-1}(s_i, \alpha_i) = i + \alpha = \beta \tag{2}$$

We can analyze the associated computational model, for this purpose the following operators are defined:

***2-tuple linguistic comparison operators***. Given two 2-tuple language values $(s_n, \alpha_1)$ and $(s_m, \alpha_2)$ representing amounts of information:

- If $n < m$, then $(s_n, \alpha_1)$ is less than $(s_m, \alpha_2)$.
- If $n = m$, then
  - (a) If $\alpha_1 = \alpha_2$, then $(s_n, \alpha_1)$ and $(s_m, \alpha_2)$ represent the same information.
  - (b) If $\alpha_1 < \alpha_2$, then $(s_n, \alpha_1)$ is less than $(s_m, \alpha_2)$.
  - (c) If $\alpha_1 > \alpha_2$, then $(s_n, \alpha_1)$ is greater than $(s_m, \alpha_2)$.

***Negation operator of a 2-tuple linguistic value.*** It is defined as:

$$neg(s_i, \alpha) = \Delta_S \left( g - (\Delta_S^{-1}(s_i, \alpha_i)) \right) \tag{3}$$

where $p + 1$ is the cardinality of the set $S$.

***Aggregation operators for 2-tuple linguistic values***. The aggregation operation used in our model are depicted below:

**Definition 4.** *Let* $((s_1, \alpha_1), \ldots, (s_p, \alpha_p))$ *be a set of 2-tuple linguistic in* $\langle S \rangle$, *and* $\omega = (\omega_1, \ldots, \omega_p)$ *be their associated weights, such that* $\sum_1^p \omega_i = 1$, *then the 2-tuple weighted average is given by the function* $F^\omega \langle S \rangle^p : \to \langle S \rangle$ *defined as:*

$$F^\omega((s_1, \alpha_1), \ldots, (s_p, \alpha_p)) = \Delta_S \left( \sum_1^p \omega_i \, \Delta_S^{-1}(s_i, \alpha_i) \right) \tag{4}$$

### 3.2. AHP Method

In an everyday environment, and more so in the business world, the problem of decision making is critical. Every day, complex problems are presented that are not easy to solve because they involve a large number of criteria, sub-criteria and alternatives [36].

Factors to be taken into account in the decision making (TD) process are the number of criteria, the decision environment and the number of experts [37], Figure 4.

- The number of criteria. If the number of criteria is greater than one, we are faced with a multi-criteria decision making (MCDM) problem. The MCDM problems are much more complicated to solve than problems involving a single criterion, because they require a step of information unification, and in many cases this information is heterogeneous.
- The decision environment. If we know exactly all the factors involved in the decision problem, we are talking about an environment of certainty. On the other hand, if the information available to us is imprecise or not very specific, we are talking about a decision problem with uncertainty. Moreover, if any of the factors responds to chance, the environment is one of risk.
- The number of experts. In the case of several experts participating in the decision making, the problem becomes more complicated; we must be able to aggregate the information from all the experts to solve the problem. However, different points of view provide the problem with a more satisfactory solution—it is known as group decision making (TDG).

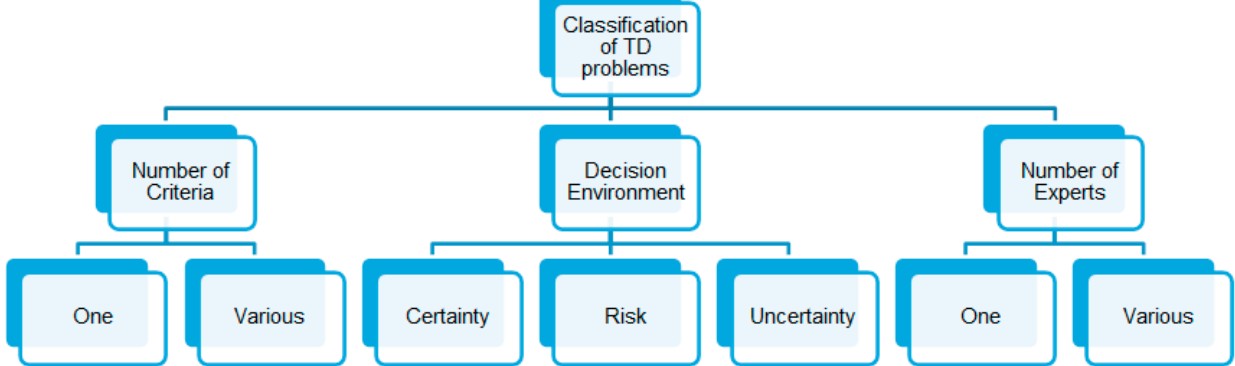

**Figure 4.** Classification of TD Problems.

Most commonly in enterprise environments, TD problems involve multiple criteria and multiple experts (MCDM-ME).

Among the MCDM models is the AHP method [15]. Based on mathematics and psychology, it is designed to solve complex multi-criteria problems [38].

The main feature of the AHP model is that the decision problem is modelled according to a hierarchy of characteristics. At the first level is the objective to be achieved, at the second level, the criteria, and sub-criteria, where the weights of each of these criteria can be determined in relation to the rest (pairwise comparison), and finally, each criterion is compared with the alternatives available to us. It is possible to determine with absolute precision and in our model, in a dynamic way, the preponderance of one alternative over another in the decision problem. Finally, once the contribution of each element to the elements of the next higher level of the hierarchy has been evaluated, an additive aggregation approach is used to calculate the overall contribution of each alternative towards achieving the primary objective [39,40].

The VIUE model will help us to determine the weights of each of the criteria that will determine the final assessment and consequently the prioritization of the interaction between customer and brand.

This whole process is detailed in the following subsections.

### 3.2.1. Structuring the Decision Model in a Hierarchical Process

The AHP method begins by structuring the decision problem as a hierarchy. The method involves breaking down the decision problem into elements based on their common characteristics and constructing a hierarchical model of various interrelated criteria to

facilitate understanding and evaluation. The highest level of the problem hierarchy features the objective (Target), while the second level includes a set of criteria $C = \{c_1, \ldots, c_{\#c}\}$, that can be further subdivided into sub-criteria $c_{ij}$, $c_{1j} = \{c_{11}, \ldots, c_{1\#C1}\}$ and so on recursively, the final level of the hierarchy consists of the alternatives $A = \{a_1, \ldots, a_{1\#A}\}$, Figure 5.

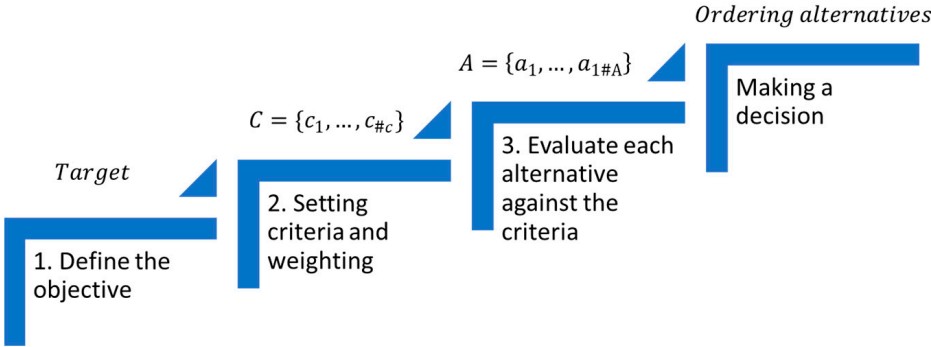

**Figure 5.** Analytic Hierarchy Process.

The decision-making objective is at the top level, the criteria and sub-criteria define the lower levels, the alternatives are defined according to criteria and sub-criteria. It is important that there are no dependencies between criteria so that the AHP methodology can be representative, and the conclusions obtained are the most appropriate for the defined decision problem.

### 3.2.2. Setting Criteria and Weighting

The opinions of the decision-makers are used to make pairwise comparisons, i.e., comparing the elements of a certain level with respect to a specific element of the next higher level. The resulting pairwise comparison matrix, $PW = (pw_{ij})\ n \times n$, where $pw_{ij}$ represents the importance of criterion $i$ relative to criterion $j$, as evaluated by the decision-makers. The matrix entries are determined by a predefined rating scale of numbers, as shown in Table 3. The matrix entries, $a_{ij}$, are governed by the following rules:

$$pw_{ij} > 0;\ pw_{ij} = \frac{1}{pw_{ij}}\ \text{and its reciprocal};\ pw_{ij} = 1\ \text{for all } i.$$

**Table 3.** Saaty Scale [41].

| Degree of Importance | Definition | Description |
|---|---|---|
| 1 | Equal importance | Equal weighting between the two criteria $i, j$. |
| 3 | Moderate importance | The weighting of criterion $i$, is moderately higher than the weighting of criterion $j$. |
| 5 | Strong importance | The weighting of criterion $i$ is higher than the weighting of criterion $j$. |
| 7 | Very strong importance | The weighting of criterion $i$ is very strong than the weighting of criterion $j$. |
| 9 | Extremely importance | The weighting of criterion $i$ is extremely strong than the weighting of criterion $j$. |
| 2, 4, 6, 8 | Intermediate values | Intermediate weighting of criteria. |
| Reciprocals | | The inverse correspondence between $i$ and $j$ can be established, according to the above specifications. |

The vector of criteria weights, $w$, is constructed using the eigenvector method, using the equation:

$$\sum_{j=1}^{n} pw_{ij}\omega_j = \lambda_{max} \times \omega_i \tag{5}$$

where $\lambda_{max}$ is the maximum eigenvalue of $PW$ and $w$ is the normalized eigenvector associated with the principal eigenvalue of $PW$.

The consistency of the AHP method can be verified through the Consistency Ratio ($CR$) which is defined as:

$$CR = CI/RI \tag{6}$$

In other words, the quotient between the Consistency Index (*CI*), defined as $\frac{\lambda_{max}-n}{n-1}$ and the Random Consistency Index (*RI*), see Table 4, which represents the consistency of a randomly generated pairwise comparison matrix.

**Table 4.** Random consistency values [41].

| *n* | 1 | 2 | 3 | 4 | 5 | 6 | 7 | 8 | 9 | 10 |
|---|---|---|---|---|---|---|---|---|---|---|
| **Random Consistency Index (*RI*)** | 0.00 | 0.00 | 0.58 | 0.9 | 1.12 | 1.24 | 1.32 | 1.41 | 1.45 | 1.49 |

If *CR* $\leq$ Consistency limits, Table 5, the results of the individual hierarchical type are satisfied and consistency is ensured, otherwise the values of the pairwise comparison items will need to be adjusted, and the judgments will need to be adjusted again by the decision-makers until they are consistent.

**Table 5.** Consistency limits [41].

| Size of the Consistency Matrix | Consistency Ratio |
|---|---|
| 3 | 5% |
| 4 | 9% |
| $\geq$5 | 10% |

### 3.2.3. Evaluate Each Alternative against the Criteria

In the same way that we have proceeded recursively to obtain the weighting of criteria and sub-criteria, we can work with the alternatives, relating each of them to each criterion. The same comparison matrix would be obtained, assigning weights to each of the alternatives according to the criterion to be compared. The model would thus obtain a matrix of weights for each alternative related to each criterion, in this case the comparison matrix $W = \left( pw_{ij} \right) n \times n$, would represent the comparison of each alternative with each other, related to each criterion. The rest of the process is the same as the one detailed in the previous point, so that a ranking of each of the alternatives could be obtained, according to the established weightings.

### 3.2.4. Making a Decision

Finally, we would rank the different alternatives and make the most appropriate decision in response to the research objective.

### 3.2.5. Sensitivity Analysis

In a decision-making process it is important to visualize and analyze the sensitivity of the result obtained, the order of the alternatives with respect to possible changes in the importance of the criteria. In sensitivity analysis, the values of the decision matrix are varied to see how the relative weights of the criteria and alternatives change. To do this, different techniques can be used, such as varying the values of the decision matrix over a specific range or introducing random errors in the values of the decision matrix [42].

### 3.3. Treatment of Heterogeneous Information

In the present work, we will perform the unification of heterogeneous information based on a 2-tuple linguistic information domain [14]. Before performing the unification process, it will be necessary to define the Basic Set of Linguistic Terms (CBTL), and the computation and results obtained will be performed on this model.

Selection of the CBTL domain $\overline{S} = \left\{ s_0, \ldots, s_p \right\}$ is performed by obtaining the set of linguistic terms of maximum granularity within the heterogeneous frame [43]. By making such a selection, we maintain the maximum degree of information represented within the linguistic domain. Once the CBTL has been selected, we go on to perform the transformation of the different expression domains to the selected CBTL set.

Information can be represented in different domains: numerical, interval and linguistic; for each of them we analyze how the process works [14], Figure 6.

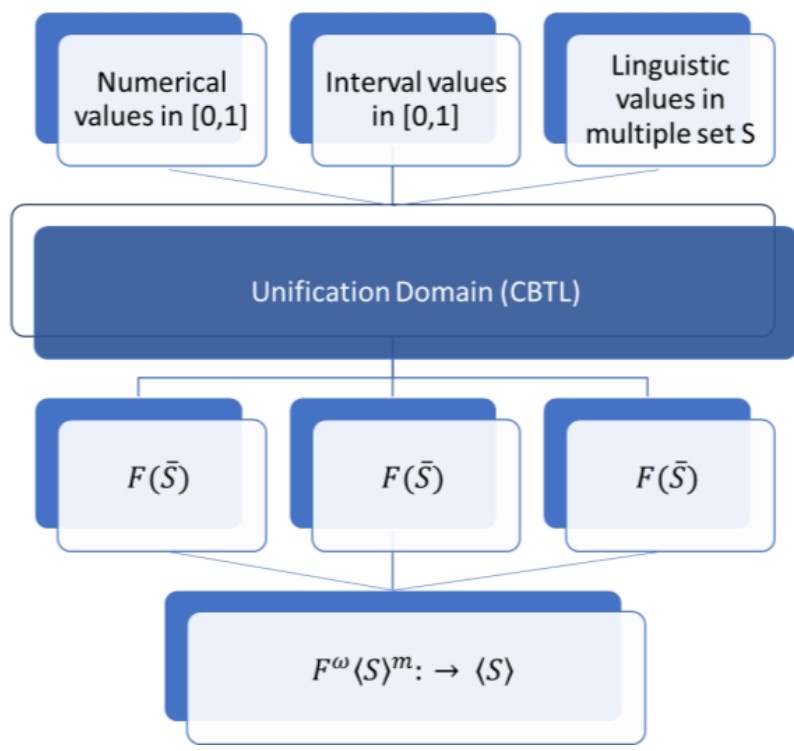

**Figure 6.** Unification model in 2-tuple linguistic [14].

3.3.1. Numerical Domain

**Definition 5.** *Given a numerical value* $n \in [0,1]$ *and the set belonging to the CBTL domain* $\overline{S} = \{\overline{s}_0, \ldots, \overline{s}_p\}$ *the numerical transformation function* $T_{N\overline{S}} : [0,1] \rightarrow F(\overline{S})$, *is defined as:*

$$T_{N\overline{S}}(n) = \{(\overline{s}_0, \gamma_0), \ldots, (\overline{s}_p, \gamma_p)\}, \overline{s}_i \in \overline{S} \tag{7}$$

With,

$$\gamma_i = \mu_{\overline{s}_i}(n) = \begin{cases} 0 \text{ if } n \notin support(\mu_{\overline{s}_i}(x)) \\ \frac{n-a_i}{b_i-a_i} \text{ if } a_i \leq n \leq b_i \\ \frac{c_i-n}{c_i-b_i} \text{ if } b_i \leq n \leq c_i \end{cases} \tag{8}$$

where $\gamma_i = \mu_{\overline{s}_i}(n) \in [0, 1]$ is the degree of association of $n$ a $\overline{s}_i \in \overline{S}$.

3.3.2. Interval Domain

**Definition 6.** *Given a value* $u = [a, b] \in P([0,1])$ *and the set belonging to the CBTL domain,* $\overline{S} = \{\overline{s}_0, \ldots, \overline{s}_p\}$ *the interval transformation function* $T_{I\overline{S}} : P([0,1]) \rightarrow F(\overline{S})$ *is defined as:*

$$T_{I\overline{S}}(u) = \left\{ \left(\overline{s}_k, \gamma_k^i\right)/k \in \{0,\ldots,p\} \right\} \tag{9}$$

where $\gamma_k^i = max_y min\{\mu_I(y), \mu_{\overline{s}_k}(y)\}$; $\mu_I(y), \mu_{\overline{s}_k}(y)$ identify, respectively, the membership functions associated with the interval $I$ and the terms $\overline{s}_k$.

$$\mu_I(y) = \begin{cases} 0 \text{ si } y < a \\ 1 \text{ si } a \leq y \leq b; \ y \in [0,1] \\ 0 \text{ si } y > b \end{cases} \tag{10}$$

### 3.3.3. Linguistic Domain

**Definition 7.** *Let $S = \{l_0, \ldots, l_h\}$, and the set belonging to the CBTL domain $\overline{S} = \{\overline{s}_0, \ldots, \overline{s}_p\}$, both sets of linguistic terms, such that $p \geq h$. The linguistic transformation function $T_{S\overline{S}} : S \rightarrow F(\overline{S})$ is defined as:*

$$T_{S\overline{S}}(l_i) = \left\{ \left( \overline{s}_k, \gamma_k^i \right) / k \in \{0, \ldots, p\} \right\} \; \forall l_i \in S \tag{11}$$

where $\gamma_k^i = max_y min \{\mu_{l_i}(y), \mu_{\overline{s}_k}(y)\}$, $i = 0, \ldots, p$ and $\mu_{l_i}(y)$, $\mu_{\overline{s}_k}(y)$ identify the membership functions that correspond to each term $l_i$, $\overline{s}_k$.

Once the heterogeneous information has been unified into a 2-tuple linguistic domain, the operations related to the LD2T domain can be applied. The results obtained in this sense are interpretable and unify in a single domain the heterogeneous evaluations in relation to a given criterion.

## 4. VIUE, Proposed Model

The Contact Center should seek to maximize customer satisfaction, minimizing costs, contributing to enriching the customer profile, its digital footprint, leading to automatic decision-making processes, Figure 7.

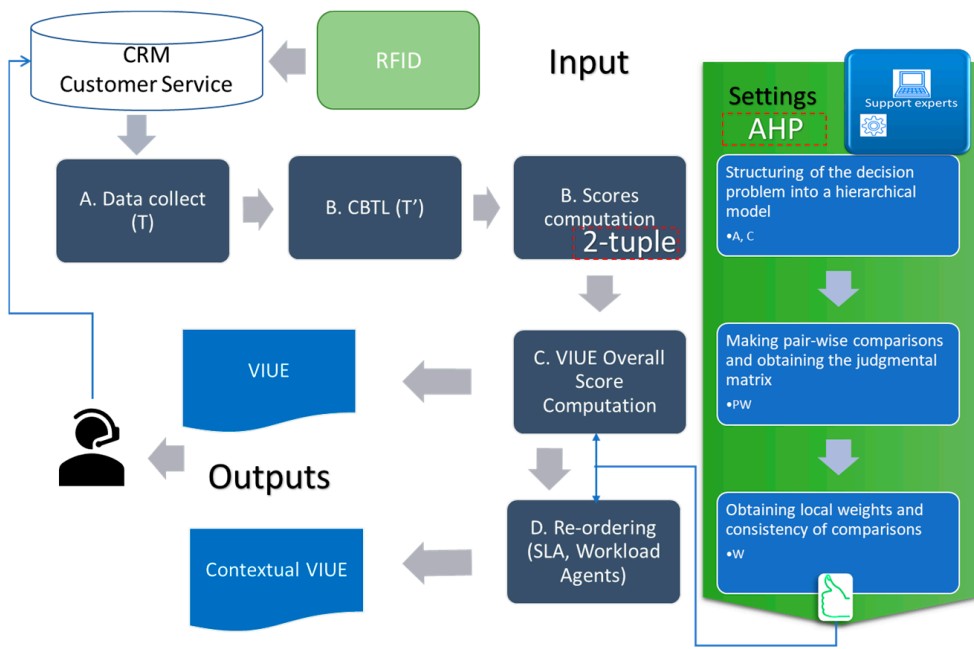

**Figure 7.** VIUE Model.

The new VIUE model proposed in this paper is based on the prioritization and personalization of customer interactions with the Contact Center. For this purpose, the interactions received in the Contact Center in real time, recorded in the CRM, are analyzed and classified according to factors such as customer service Value ($V$), Impact ($I$), Urgency ($U$) and the Emotional nature of the interaction ($E$); therefore, the priority of interaction can be defined as an aggregate value of all the above criteria $\mathbb{P} = f(V, I, U, E)$. The above criteria will provide a first classification of the interactions and are based on the ITIL methodology [10,11], which recommends determining the priority of incident resolution based on the impact and urgency of the interactions. For our Customer Service model, we have extended this methodology with concepts such as the value of the customer in their relationship with the Contact Center [7], as well as the emotional nature of such interaction [12], all of which directly affect the processes and quality of service.

Second, this classification is alive. We are continuously receiving interactions in real time and, therefore, the Contact Center activity queue undergoes modifications. It may

happen that, if we only classify the interactions by these criteria, we could leave customers with a low priority rating, and they remain unattended. To avoid this, a second contextual classification is made, which depends on the SLAs defined for the customer or customer segment, i.e., the maximum waiting time ($T$) and the workload of the Contact Center ($C$) is what we have called Contextual VIUE.

The model's parameters are determined based on the information provided by the customer service experts. Because it is a dynamic model, the expert staff, according to stationarity, history, workload, and other factors depending on the sector of activity, will be able to balance in real time the weights of each criterion. In addition, they will oversee determining and applying final treatment and personalization strategies with each customer or group of customers. In addition, they will analyze if after the first VIUE classification, it is enough to determine the priority of attention of the customers, without the need to apply the contextual adjustment of waiting time and workload.

The process to be followed is as follows:

- CRM data collection.
- Determine the CBTL expression domain for each criterion.
- We apply the 2-tuple model on the data obtained in the previous step.
- We obtain the global valuation of each interaction by applying the AHP model.
- If necessary, we establish a reordering of priorities according to the SLA and workload values of the Contact Center, thus applying an adjustment to the model.

The proposed model is explained in more detail below.

### 4.1. Data Collection

In CRM terminology, the incident represented by a ticket or case corresponds to any type of customer service request or complaint, this type of incident occurs after the sales process. For our purpose, the incident is automatically recorded in the CRM by the Contact Center management system. For this, it will be necessary that the customer is attended by a Bot or a human agent (call, chat). During this process of attention, relevant information will be collected that will categorize the ticket and thus trigger the process to define the priority and personalization of care. In this case, the data set T will be identified by the following parameters, some of them calculated and others expressed in a 2-tuple domain. In the CRM, the RFID rating of each customer is collected, therefore. Given,

$$T = \{(u_i, RFID_i, ticket\_id_i, ticket\_date_i, trouble\_id_i, ticket\_impact_i, ticket\_urgency_i, ticket\_emotion_i)\}$$

Representing the detail of each request, where for each customer $u_i$ we have:

- $RFID_i$: represents the customer's value from the perspective of the Contact Center. For the case at hand, it is defined on a linguistic scale in a 2-tuple domain.
- $ticket\_id_i$: corresponds to the code that uniquely identifies each ticket, i.e., an incident opened by the customer $u_i$, with $i \in 1, \ldots, \#T$.
- $ticket\_date_i$: corresponds to the date when the service was originally required.
- $trouble\_id_i$: identifies the type of request, complaint, or problem the customer is having.
- $ticket\_impact_i$: ticket relevance is a standard feature of most CRMs. This variable is responsible for measuring the effects of the ticket on business processes. It is generally expressed on an ordinal and/or linguistic scale of n values, so that the higher the value, greater relevance of the ticket. In this article, and considering the use case, the scale used will consist of five values {very low, low, moderate, high, very high}. As this is a linguistic scale, we will consider its modeling with the set of $S$.
- $ticket\_urgency_i$: most CRMs include ticket urgency as a standard feature. It is a measure of how much damage the issue can do to the business. It is usually expressed in the same way as the impact on an ordinal and/or linguistic scale of n values. In this report, and considering the use case worked on, we will consider the scale to have five values {very low, low, moderate, high, very high}. As this is a linguistic scale, we will consider its modeling with the set of $S$.

- *ticket_emotion$_i$*: corresponds to the emotional value of the interaction, it is a measure of the "degree of anger" of the customer in his interaction with the brand. For which we will perform a sentiment analysis that will allow us to classify the interaction and the emotional nature of the interaction [22]. The sentiment analysis will be carried out, considering the use case worked on, to a fuzzy model with three values {low, moderate, high}.

*4.2. CBTL Domain, Scores Computation*

In this step, the 2-tuple scores are obtained for the set $VIUE = \{u_1, V_1, I_1, U_1, E_1 \ldots, u_{\#U}, V_{\#U}, I_{\#U}, U_{\#U}, E_{\#U}\}$.

Therefore, we must calculate the following variables: $V_e, I_e, U_e, E_e \in S \times [-0.5, 0.5)$. For each customer $u_e$, we obtain $A_e = (A_{e1}, A_{e2}, A_{e3}, A_{e4})$ with $A_{e1} = V_e$, $A_{e2} = I_e$, $A_{e3} = U_e$, $A_{e4} = E_e$.

Variables 1, 2 and 3 ($V$, $I$, $U$) are defined in a linguistic domain $S5$, variable 4 ($E$) is defined in a linguistic domain $S3$, $T_{S\overline{S}} : S \rightarrow F(\overline{S})$. Thus, we need to apply the domain transformation of $S3$ to $S5$ according to the Equation (11). In this way we will have the variables $A_{ei}$ in the same linguistic domain $S5$.

Once all the variables have been unified into a CBTL fuzzy domain, we will transform this domain into 2-tuple linguistic variables and thus operate on each of the values through the 2-tuple computational model.

*4.3. VIUE, Overall Score*

In this step the value of 2-tuple $VIUE_e$, that characterizes the score and priority $P_e$ of each interaction with the Contact Center $V_e, I_e, U_e, E_e$, is calculated for each customer using Equation (4), in such a way that $P_e = VIUE_e = F^\omega[A_{ei}]$.

We will structure the decision problem in a hierarchical model (AHP), and then elaborate the pairwise comparison matrix, Equation (5), obtaining the vector of weights for each of the variables, $W = w_V, w_I, w_U, w_E$.

*4.4. Contextual VIUE, Reordering*

In a first phase, we can stay with the sorting of tickets determined by the previous steps, however, one more action is proposed. As the prioritization process is dynamic, the priorities are updated in real time as new incidents continue to enter. In certain business environments, it is necessary to deepen this first classification. Some customers could remain in queue, without being attended. Consequently, a possible improvement to the model consists of, once having determined the value of the priority $P_e$, with the phases described above, perform a new reordering that responds to the waiting time. In this way, when a % of the waiting time value marked in the SLA is reached, the interaction will go to the head in the queue of interactions. Alternatively, the workload of the Contact Center also influences, or in inverse terminology, for the work capacity of the Contact Center, the higher the workload, the lower the capacity and vice versa. The Contact Center manager will define the minimum and maximum workload values, as well as the maximum waiting time value per customer or customer category.

Additionally, we will perform a customization process in the interaction, considering the previous criteria: initial priority obtained, waiting time and the workload of the team of agents assigned to resolve the incident, because these, in this case, have additional tasks assigned to them in addition to incident handling. The proposed alternatives are:

- Alternative 1. The interaction will be attended by a Bot in the corresponding channel.
- Alternative 2. The interaction will be attended by generalist personnel.
- Alternative 3. The interaction will be attended by specialized personnel.

The model used in this new reordering will be like the one described in the previous steps, and will be based on:

- Definition of the CBTL domain, in this case, the criteria are defined as follows: priority $P$, obtained from the overall score VIUE model, described in the previous phase, which

is in a fuzzy domain $F(\overline{S})$ = {very low, low, medium, high, very high}; waiting time ($T$) and workload ($C$) are in a numerical domain $T_{N\overline{S}} : [0,1] \rightarrow F(\overline{S})$, Equation (7).

- Unification of heterogeneous information to the defined CBTL domain ($S5$).
- Evaluate the weights of each criterion involved in the decision-making process $W = w_P, w_T, w_C$.
- Prioritization and recommendations for customization of interactions according to the weights of each criterion and the overall rating of each interaction obtained as a function of the criteria priority, waiting time and workload.

The final domain obtained, Contextual VIUE, $CVIUE_i \in S \times [-0.5, 0.5)$, is represented in a 2-tuple value, thus establishing a reordering of all interactions according to the criteria seen in the previous points.

The process described above provides us with a completely dynamic order of attention to incidents, because as new incidents come in, priorities are adjusted according to the waiting time and work capacity of the Contact Center, and on the other hand, the process allows us to recommend personalization actions in the interaction.

Let $U = \{u_1, \ldots, u_{\#U}\}$ the customers set; the 2-tuple scores for the VIUE set are defined as, $CVIUE = \{P_1, T_1, C_1, \ldots, P_{\#U}, T_{\#U}, C_{\#U}\}$.

For each customer $u_e$, $e = 1, \ldots, \#U$, we obtain $A_e = (A_{e1}, A_{e2}, A_{e3})$ with $A_{e1} = P_e$, $A_{e2} = T_e$, $A_{e3} = C_e$. It defines, $rank \in \{1, \ldots, \#U\}$ and classification of each customer with respect to each of these variables:

$$percent\_rank_e(B_{ei}) = \frac{rank(B_{ei} - 1)}{\#U - 1} \tag{12}$$

with $percent\_rank \in [0,1]$, $e = 1, \ldots, \#U$ and $i = 1, \ldots, 3$. The final score 2-tuple $A_{ei}$ is calculated using the following method:

$$A_{ei} = \begin{cases} \Delta(percent\_rank(B_{ei})), if\ i = 2 \\ B_{ei},\ if\ i = 1 \\ neg(\Delta(percent\_rank(B_{ei}), if\ i = 3 \end{cases} \tag{13}$$

where $\Delta(\cdot)$ and $neg(\cdot)$ have been defined in Equations (1) and (3) respectively, and *percent_rank* in Equation (12). In this case we use the negation function with the workload value, as higher scores correspond to higher workload in the Contact Center and, consequently, lower responsiveness.

## 5. VIUE Model, Practical Application

This section provides an illustrative example of how the new VIUE model can be applied, adapted to a software licensing manufacturer, which collaborates with technology partners that distribute and implement the manufacturer's solutions, in a model Business to Business (B2B).

### 5.1. Data Collection

The ticket information is managed in an operational CRM, in this case Salesforce, its structure is based on the model shown in Figure 8.

The set of $T$ tickets is available, corresponding to real-time interactions between partner and manufacturer; this set of tickets is managed from the internal CRM tool. On the other hand, for each customer we have its valuation based on the history of relations with the Contact Center, RFID model.

The interaction between partner and manufacturer is initiated through a Bot, either by Phone or Chat. The Bot oversees making an initial assessment of factors such as emotional, impact and urgency of the interaction.

For the emotional factor, a sentiment analysis of the interaction is made, in any of the communication channels [44,45]. This results in a score in a linguistic domain $S = \{low, moderate, high\}$.

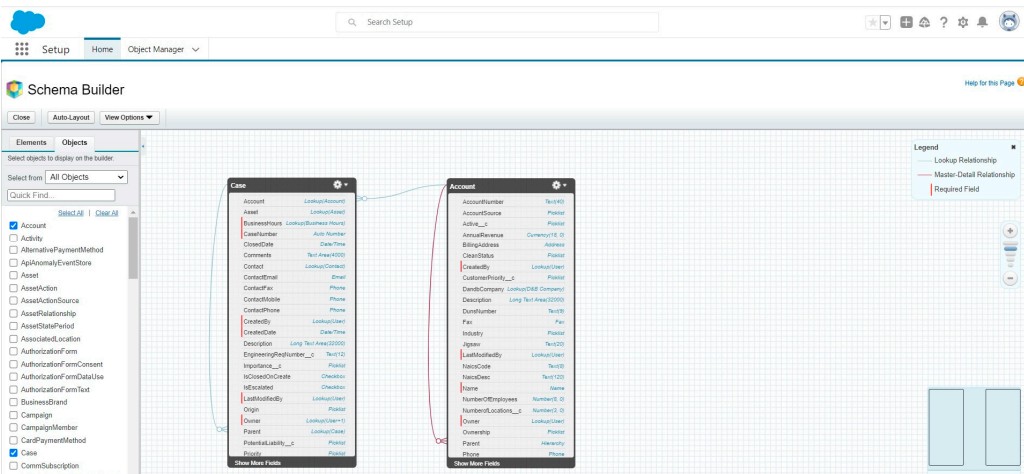

**Figure 8.** Salesforce CRM. Schema Builder–Service Setup.

Figure 9 shows the number of incidents by month in the year 2020.

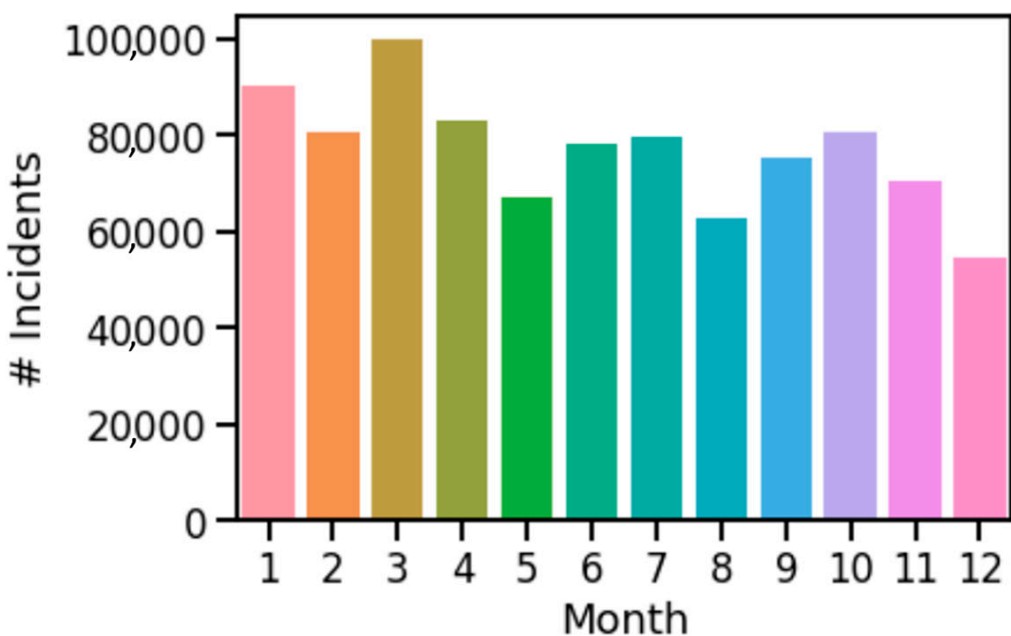

**Figure 9.** Number (#) of incidents by month in 2020.

The analysis of the quality department of the software manufacturer determines, in real time through a custom developed program and according to certain KPIs specific to the business, the degree of impact and urgency of any incident, using the ITIL methodology [10,11]. A linguistic scale is thus established $\overline{S} = \{very\ low,\ low,\ moderate,\ high,\ very\ high\}$ associated with impact and urgency. The type of company we are analyzing, the impact is related to the number of people (and their importance) affected by the incident. For example, a Cloud System downtime causes a very high impact with many customers affected and strong effects on business processes. The urgency is about how long it will take before the impact on the business is significant; for this service it also causes a very high urgency, with very high economic losses, due to the SLAs and the corresponding penalties, in addition to the loss of confidence on the part of the customer/partner.

*5.2. CBTL Domain, Scores Computation*

Through the selection of the CBTL domain, we can obtain a first approximation to the 2-tuple linguistic model, with the ratings obtained for the customer value from the RFID

model, and the linguistic values in 2-tuple format of the other three criteria, emotional, impact and urgency.

We have selected a set of interactions in the same time range and corresponding to a total of 25 customers [7,46].

The criteria customer value, impact and urgency are represented under a linguistic domain $S5$, the emotion criterion is defined in a linguistic domain $S3$, $T_{S\overline{S}} : S \rightarrow F(\overline{S})$. The domain transformation from $S3$ to $S5$ is applied according to Equation (11). This obtains the following valuations shown in Table 6.

**Table 6.** VIUE, Matrix in the $S5$ Domain.

| Ticket_ID | u | V = RFID | I | U | E |
|---|---|---|---|---|---|
| 91 | wRlM0 | (L, 0.120) | VL | VH | M |
| 113 | wRlth | (L, 0.027) | L | VL | (L, 0.33) |
| 135 | wRmUk | (L, 0.089) | M | H | M |
| 136 | wRmWM | (L, 0.089) | VH | VH | (L, 0.33) |
| 197 | wRqQO | (L, 0.090) | VL | H | (H, 0.33) |
| 33 | wRiEY | (VH, −0.299) | H | VL | (L, 0.33) |
| 45 | wRiro | (VH, −0.200) | M | L | (L, 0.33) |
| 71 | wRjqc | (VH, −0.267) | H | M | (H, 0.33) |
| 102 | wRljY | (VH, −0.382) | M | VH | (H, 0.33) |
| 104 | wRllZ | (VH, −0.424) | M | VH | (H, 0.33) |
| 6 | wRf6k | (M, −0.398) | VL | VH | (L, 0.33) |
| 65 | wRjYB | (M, −0.425) | H | L | (L, 0.33) |
| 214 | wRrLB | (M, −0.394) | L | L | (H, 0.33) |
| 310 | wRyOF | (M, −0.436) | M | L | (H, 0.33) |
| 317 | wRyni | (M, −0.436) | VH | M | (H, 0.33) |
| 1 | wRenT | (M, 0.445) | H | M | M |
| 38 | wRiOQ | (M, 0.429) | VL | H | (L, 0.33) |
| 44 | wRipv | (H, −0.028) | M | VL | (L, 0.33) |
| 47 | wRish | (M, 0.454) | L | H | (L, 0.33) |
| 56 | wRiyz | (H, −0.418) | VH | M | (L, 0.33) |
| 5 | wRf5G | (M, 0.498) | VL | M | M |
| 34 | wRiIG | (M, 0.434) | L | VL | M |
| 35 | wRiIV | (M, 0.392) | VL | VL | (H, 0.33) |
| 41 | wRinp | (H, −0.007) | M | M | (H, 0.33) |
| 54 | wRiwj | (M, 0.396) | VL | M | M |

### 5.3. VIUE, Overall Score

In this stage, the importance of each of the features of the VIUE model must be obtained before calculating the overall interaction score. For this purpose, as mentioned above, the AHP model will be used.

Using the Saaty scale, the consulted expert has provided the following matrix (Table 3):

$$\text{PW} = \begin{bmatrix} & V & I & U & E \\ V & 1 & 1/3 & 1/3 & 3 \\ I & 3 & 1 & 1 & 5 \\ U & 3 & 1 & 1 & 5 \\ E & 1/3 & 1/5 & 1/5 & 1 \end{bmatrix}$$

Only when $CR \leq 0.09$ (see Equation (6)) are the results of the individual hierarchical type satisfied and consistency is guaranteed. In this case, $CR = 0.016$ so the results are valid for the model.

The final weights calculated are $W = \{w_V = 0.153, w_I = 0.389, w_U = 0.389, w_E = 0.069\}$. Therefore, the expert has given greater relevance to the impact of incidents, followed by the urgency and the emotional nature of the interaction. It is true that in B2B models the emotional nature of the interaction is important, but not as much as in B2C models. As relationships in B2B models are long-term, the commitment between manufacturer and

distributor/software implementer is long-lasting, not short-term. The bet that a partner makes to train its staff and grow together with the manufacturer is very high; the business model is based on the manufacturer's products and an incidence with high emotional valuation, should not be a determining factor in the prioritization of such interaction [46]. Table 7 shows the overall scores of the VIUE model.

**Table 7.** VIUE, Overall Score.

| Ticket_ID | u | V = RFID | I | U | E | P = VIUE |
|---|---|---|---|---|---|---|
| 91 | wRlM0 | (L, 0.120) | VL | VH | M | (L, −0.135) |
| 113 | wRlth | (L, 0.027) | L | VL | (L, 0.33) | (L, −0.362) |
| 135 | wRmUk | (L, 0.089) | M | H | M | (M, 0.249) |
| 136 | wRmWM | (L, 0.089) | VH | VH | (L, 0.33) | (H, 0.370) |
| 197 | wRqQO | (L, 0.090) | VL | H | (H, 0.33) | (L, −0.437) |
| 33 | wRiEY | (VH, −0.299) | H | VL | (L, 0.33) | (L, −0.174) |
| 45 | wRiro | (VH, −0.200) | M | L | (L, 0.33) | (M, −0.159) |
| 71 | wRjqc | (VH, −0.267) | H | M | (H, 0.33) | (H, −0.254) |
| 102 | wRljY | (VH, −0.382) | M | VH | (H, 0.33) | (H, 0.118) |
| 104 | wRllZ | (VH, −0.424) | M | VH | (H, 0.33) | (H, 0.111) |
| 6 | wRf6k | (M, −0.398) | VL | VH | (L, 0.33) | (M, −0.107) |
| 65 | wRjYB | (M, −0.425) | H | L | (L, 0.33) | (M, −0.111) |
| 214 | wRrLB | (M, −0.394) | L | L | (H, 0.33) | (L, 0.253) |
| 310 | wRyOF | (M, −0.436) | M | L | (H, 0.33) | (M, −0.364) |
| 317 | wRyni | (M, −0.436) | VH | M | (H, 0.33) | (H, −0.198) |
| 1 | wRenT | (M, 0.445) | H | M | M | (M, 0.457) |
| 38 | wRiOQ | (M, 0.429) | VL | H | (L, 0.33) | (L, 0.369) |
| 44 | wRipv | (H, −0.028) | M | VL | (L, 0.33) | (L, 0.325) |
| 47 | wRish | (M, 0.454) | L | H | (L, 0.33) | (M, 0.024) |
| 56 | wRiyz | (H, −0.418) | VH | M | (L, 0.33) | (H, −0.179) |
| 5 | wRf5G | (M, 0.498) | VL | M | M | (L, 0.299) |
| 34 | wRiIG | (M, 0.434) | L | VL | M | (L, −0.100) |
| 35 | wRiIV | (M, 0.392) | VL | VL | (H, 0.33) | (L, −0.404) |
| 41 | wRinp | (H, −0.007) | M | M | (H, 0.33) | (M, 0.244) |
| 54 | wRiwj | (M, 0.396) | VL | M | M | (L, 0.283) |

*5.4. Contextual VIUE, Reordering*

In the proposed use case, severe SLAs are defined with each partner and the response and incident resolution times are critical. For this reason, we will apply the contextual VIUE model adjustment. Thus, once the priority value has been determined with the phases described above, we will carry out a reordering of the interactions. To do this, we will consider the following criteria: defined priority obtained from the model $P = VIUE$, waiting time $T$, and workload $C$, of the team of agents assigned to resolve the incident, as these, in this case, have additional tasks assigned to them in addition to attending to incidents. The steps to be obeyed are as follows:

- First, Table 8 shows how the tickets are ordered by priority in the management of the incident by applying VIUE, and then we proceed to reorder them according to the criteria expressed in the previous paragraph.
- Second, the contextual VIUE score is obtained, Table 9, based on the initial priority (VIUE), waiting time and workload of the Contact Center.
- Third, the ratings of the contextual 2-tuple VIUE set are obtained, Table 10.
- Finally, we would obtain the overall assessment ordered, Table 11, and therefore the final priority and the recommendations for customization by applying AHP.

**Table 8.** Overall VIUE Score Ordered.

| Ticket_ID | u | V = RFID | I | U | E | P = VIUE |
|---|---|---|---|---|---|---|
| 136 | wRmWM | (L, 0.089) | VH | VH | (L, 0.33) | (H, 0.370) |
| 102 | wRljY | (VH, −0.382) | M | VH | (H, 0.33) | (H, 0.118) |
| 104 | wRllZ | (VH, −0.424) | M | VH | (H, 0.33) | (H, 0.111) |
| 56 | wRiyz | (H, −0.418) | VH | M | (L, 0.33) | (H, −0.179) |
| 317 | wRyni | (M, −0.436) | VH | M | (H, 0.33) | (H, −0.198) |
| 71 | wRjqc | (VH, −0.267) | H | M | (H, 0.33) | (H, −0.254) |
| 1 | wRenT | (M, 0.445) | H | M | M | (M, 0.457) |
| 135 | wRmUk | (L, 0.089) | M | H | M | (M, 0.249) |
| 41 | wRinp | (H, −0.007) | M | M | (H, 0.33) | (M, 0.244) |
| 47 | wRish | (M, 0.454) | L | H | (L, 0.33) | (M, 0.024) |
| 6 | wRf6k | (M, −0.398) | VL | VH | (L, 0.33) | (M, −0.107) |
| 65 | wRjYB | (M, −0.425) | H | L | (L, 0.33) | (M, −0.111) |
| 45 | wRiro | (VH, −0.200) | M | L | (L, 0.33) | (M, −0.159) |
| 310 | wRyOF | (M, −0.436) | M | L | (H, 0.33) | (M, −0.364) |
| 38 | wRiOQ | (M, 0.429) | VL | H | (L, 0.33) | (L, 0.369) |
| 44 | wRipv | (H, −0.028) | M | VL | (L, 0.33) | (L, 0.325) |
| 5 | wRf5G | (M, 0.498) | VL | M | M | (L, 0.299) |
| 54 | wRiwj | (M, 0.396) | VL | M | M | (L, 0.283) |
| 214 | wRrLB | (M, −0.394) | L | L | (H, 0.33) | (L, 0.253) |
| 34 | wRiIG | (M, 0.434) | L | VL | M | (L, −0.100) |
| 91 | wRlM0 | (L, 0.120) | VL | VH | M | (L, −0.135) |
| 33 | wRiEY | (VH, −0.299) | H | VL | (L, 0.33) | (L, −0.174) |
| 113 | wRlth | (L, 0.027) | L | VL | (L, 0.33) | (L, −0.362) |
| 35 | wRiIV | (M, 0.392) | VL | VL | (H, 0.33) | (L, −0.404) |
| 197 | wRqQO | (L, 0.090) | VL | H | (H, 0.33) | (L, −0.437) |

**Table 9.** Contextual VIUE, with numeric values.

| Ticket_ID | u | P = VIUE | T | C |
|---|---|---|---|---|
| 136 | wRmWM | (H, 0.370) | 6 | 66 |
| 102 | wRljY | (H, 0.118) | 1 | 37 |
| 104 | wRllZ | (H, 0.111) | 19 | 76 |
| 56 | wRiyz | (H, −0.179) | 6 | 62 |
| 317 | wRyni | (H, −0.198) | 16 | 22 |
| 71 | wRjqc | (H, −0.254) | 16 | 72 |
| 1 | wRenT | (M, 0.457) | 16 | 96 |
| 135 | wRmUk | (M, 0.249) | 19 | 88 |
| 41 | wRinp | (M, 0.244) | 13 | 29 |
| 47 | wRish | (M, 0.024) | 5 | 85 |
| 6 | wRf6k | (M, −0.107) | 7 | 20 |
| 65 | wRjYB | (M, −0.111) | 1 | 22 |
| 45 | wRiro | (M, −0.159) | 14 | 41 |
| 310 | wRyOF | (M, −0.364) | 4 | 81 |
| 38 | wRiOQ | (L, 0.369) | 20 | 31 |
| 44 | wRipv | (L, 0.325) | 16 | 39 |
| 5 | wRf5G | (L, 0.299) | 2 | 60 |
| 54 | wRiwj | (L, 0.283) | 19 | 26 |
| 214 | wRrLB | (L, 0.253) | 0 | 41 |
| 34 | wRiIG | (L, −0.100) | 8 | 94 |
| 91 | wRlM0 | (L, −0.135) | 18 | 76 |
| 33 | wRiEY | (L, −0.174) | 4 | 94 |
| 113 | wRlth | (L, −0.362) | 9 | 66 |
| 35 | wRiIV | (L, −0.404) | 14 | 28 |
| 197 | wRqQO | (L, −0.437) | 0 | 91 |

**Table 10.** Contextual, 2-tuple VIUE assessment.

| Ticket_ID | u | P = VIUE | T | C |
| --- | --- | --- | --- | --- |
| 136 | wRmWM | (H, 0.370) | (L, 0.05) | (M, −0.075) |
| 102 | wRljY | (H, 0.118) | (VL, 0.05) | (H, 0.037) |
| 104 | wRllZ | (H, 0.111) | (VH, −0.05) | (L, 0.05) |
| 56 | wRiyz | (H, −0.179) | (L, 0.05) | (M, −0.025) |
| 317 | wRyni | (H, −0.198) | (H, 0.05) | (VH, −0.025) |
| 71 | wRjqc | (H, −0.254) | (H, 0.05) | (L, 0.1) |
| 1 | wRenT | (M, 0.457) | (H, 0.05) | (VL, 0.05) |
| 135 | wRmUk | (M, 0.249) | (VH, −0.05) | (L, −0.1) |
| 41 | wRinp | (M, 0.244) | (H, −0.1) | (VH, −0.113) |
| 47 | wRish | (M, 0.024) | L | (L, −0.062) |
| 6 | wRf6k | (M, −0.107) | (L, 0.1) | VH |
| 65 | wRjYB | (M, −0.111) | (VL, 0.05) | (VH, −0.025) |
| 45 | wRiro | (M, −0.159) | (H, −0.05) | (H, −0.012) |
| 310 | wRyOF | (M, −0.364) | (L, −0.05) | (L, −0.012) |
| 38 | wRiOQ | (L, 0.369) | VH | (H, 0.113) |
| 44 | wRipv | (L, 0.325) | (H, 0.05) | (H, 0.012) |
| 5 | wRf5G | (L, 0.299) | (VL, 0.1) | M |
| 54 | wRiwj | (L, 0.283) | (VH, −0.05) | (VH, −0.075) |
| 214 | wRrLB | (L, 0.253) | VL | (H, −0.012) |
| 34 | wRiIG | (L, −0.100) | (M, −0.1) | (VL, 0.075) |
| 91 | wRlM0 | (L, −0.135) | (VH, −0.1) | (L, 0.05) |
| 33 | wRiEY | (L, −0.174) | (L, −0.05) | (VL, 0.075) |
| 113 | wRlth | (L, −0.362) | (M, −0.05) | (M, −0.075) |
| 35 | wRiIV | (L, −0.404) | (H, −0.05) | (VH, −0.1) |
| 197 | wRqQO | (L, −0.437) | VL | (VL, 0.113) |

**Table 11.** Contextual VIUE Overall Score Ordered.

| Ticket_ID | u | P = VIUE | T | C | Contextual VIUE |
| --- | --- | --- | --- | --- | --- |
| 38 | wRiOQ | (L, 0.369) | VH | (H, 0.113) | (VH, −0.355) |
| 54 | wRiwj | (L, 0.283) | (VH, −0.05) | (VH, −0.075) | (VH, −0.469) |
| 135 | wRmUk | (M, 0.249) | (VH, −0.05) | (L, −0.1) | (H, 0.190) |
| 41 | wRinp | (M, 0.244) | (H, −0.1) | (VH, −0.113) | (H, 0.178) |
| 104 | wRllZ | (H, 0.111) | (VH, −0.05) | (L, 0.05) | (H, 0.140) |
| 35 | wRiIV | (L, −0.404) | (H, −0.05) | (VH, −0.1) | (H, 0.016) |
| 317 | wRyni | (H, −0.198) | (H, 0.05) | (VH, −0.025) | (H, −0.012) |
| 91 | wRlM0 | (L, −0.135) | (VH, −0.1) | (L, 0.05) | (H, −0.043) |
| 44 | wRipv | (L, 0.325) | (H, 0.05) | (H, 0.012) | (H, −0.051) |
| 45 | wRiro | (M, −0.159) | (H, −0.05) | (H, −0.012) | (H, −0.311) |
| 71 | wRjqc | (H, −0.254) | (H, 0.05) | (L, 0.1) | (M, 0.493) |
| 1 | wRenT | (M, 0.457) | (H, 0.05) | (VL, 0.05) | (M, 0.299) |
| 6 | wRf6k | (M, −0.107) | (L, 0.1) | VH | (M, −0.233) |
| 113 | wRlth | (L, −0.362) | (M, −0.05) | (M, −0.075) | (M, −0.268) |
| 34 | wRiIG | (L, −0.100) | (M, −0.1) | (VL, 0.075) | (M, −0.296) |
| 56 | wRiyz | (H, −0.179) | (L, 0.05) | (M, −0.025) | (M, −0.411) |
| 136 | wRmWM | (H, 0.370) | (L, 0.05) | (M, −0.075) | (L, 0.342) |
| 47 | wRish | (M, 0.024) | L | (L, −0.062) | (L, 0.321) |
| 65 | wRjYB | (M, −0.111) | (VL, 0.05) | (VH, −0.025) | (L, 0.073) |
| 310 | wRyOF | (M, −0.364) | (L, −0.05) | (L, −0.012) | (L, 0.065) |
| 214 | wRrLB | (L, 0.253) | VL | (H, −0.012) | (L, −0.123) |
| 33 | wRiEY | (L, −0.174) | (L, −0.05) | (VL, 0.075) | (L, −0.221) |
| 102 | wRljY | (H, 0.118) | (VL, 0.05) | (H, 0.037) | (L, −0.299) |
| 5 | wRf5G | (L, 0.299) | (VL, 0.1) | M | (L, −0.304) |
| 197 | wRqQO | (L, −0.437) | VL | (VL, 0.113) | (VL, 0.223) |

In this case, the priority $P = VIUE$ is in a fuzzy domain $F(\overline{S})$, $\overline{S} = \{very\ low,\ low,\ moderate,\ high,\ very\ high\}$; waiting time ($T$) and workload ($C$) are in a numerical domain $T_{N\overline{S}} : [0, 1] \rightarrow F(\overline{S})$, therefore, we will apply Equation (7).

The waiting time, in this case, is directly related to the SLA. In this model, it has been considered that interactions cannot be on hold for more than 80% of the SLA value, which is the attention time committed to each partner. As soon as an interaction exceeds 80% of the SLA value, it becomes a priority.

The Contact Center business manager has defined the maximum SLA value for waiting time at 20 min, and the Contact Center load per business area will be between 20–100% dedication.

Next, we unified the information to a CBTL domain in $S5$, and proceeded to transform the load value with the function $neg(\cdot)$, Equation (3), lower attention possibility at higher workload, or what is the same, higher responsiveness at lower workload, obtaining the following results, shown in the Table 10.

This stage involves determining the relative importance of each variable in the contextual VIUE model prior to calculating the overall score for each interaction. To accomplish this, the AHP model will be employed.

The expert consulted has specified the following matrix using the Saaty scale (Table 3):

$$\text{PW} = \begin{array}{c} \\ P \\ T \\ C \end{array} \begin{bmatrix} P & T & C \\ 1 & 1/5 & 1 \\ 5 & 1 & 3 \\ 1 & 1/3 & 1 \end{bmatrix}$$

Only when $CR \leq 0.05$ (see Equation (6)) are the results of the individual hierarchical type satisfied and consistency is guaranteed. In this case, $CR = 0.025$ so the results are valid for the model.

The final weights calculated are $W = \{w_P = 0.158,\ w_T = 0.655, w_C = 0.187\}$.

Therefore, the expert has predictably given greater relevance to the waiting time, followed by the workload of the contact center. This implies a reordering of the priority of open tickets, mainly based on waiting time and workload. We now calculate the global valuation of interactions and the new reordering considering these three factors. It is important to note that this process must be executed every time a new ticket is opened in the CRM, so that the waiting queues speed up the response to open tickets whose waiting times are longer.

The case whose Ticket id is equal to 136, goes from having a priority (H, 0.370) to a priority (L, 0.342). This is because the waiting time is low (L, 0.05) and the capacity (inverse value of the load) of the Contact Center is medium (M, −0.075).

Another example in the opposite case is the Ticket whose id is 35. As can be seen, the initial priority is low (L, −0.404), but the waiting time is high (H, −0.05) and the Contact Center capacity is very high (VH, −0.1). Finally, and after the whole process is done, the priority given to the interaction is high (H, 0.016).

On the other hand, the degree of personalization will be directly proportional to the priority of the interaction and the waiting time, and inversely proportional to the workload of the Contact Center area in charge of its attention. In other words, when the workload threshold of the area is exceeded, the partner will be attended to by seeking an auto response through a Bot or will be invited to send the incident via email, according to the alternatives indicated in the definition of the contextual VIUE model.

## 6. Discussion

In the traditional Contact Center, there is a double measurement, often prioritizing quantitative objectives, among others (first call resolution, response time, abandonment rate, calls handled, efficiency, talk time, unproductive time) over the qualitative objectives, customer satisfaction and service level. This double measure implies an emotional stress on the agent and a high turnover that results in indirect costs for the organization, and the consequent erosion of the service [47].

Traditional contact center routing and handling technologies will try to route and filter the interaction to the first available agent or to the specialist agent [48]. The management of this type of technology is limited to routing to the operator that is available or keeping the customer in a waiting queue until they are attended to. In some cases, the customer is identified in such a way that they are classified according to the customer segment weighting strategy established by the organization, CLV.

The data that have been worked on in the previous section come from a company that manufactures software solutions. As conclusions to this study applied to this company, we can confirm that the procedure designed fits with the technological change that the company wants to implement in the customer service process.

Applying the process of valuing customers in their relationship with the Contact Center, RFID, and adding criteria that are present in the interaction, emotional value, impact, and urgency, we can make a first approximation to a prioritization of tickets. For this, we used the 2-tuple model, through the use of the AHP methodology. We have been able to assign weight to each variable within the contextual VIUE model, allowing us to calculate an overall score for each interaction and create an initial ranking of the interactions.

As the process is dynamic, we must consider additional factors. The first is the customer service SLAs, in this case, the SLA defined for response time is less than 20 min. Second, the workload of the Contact Center is fixed between 20 and 80% of the total time. Using the same methodology as the one developed in the previous step, unification of all the information to a CBTL domain and 2-tuple, to later aggregate all the information using the AHP methodology that provides the weights of each of the criteria, we can obtain the global score of each interaction, and therefore its priority.

In addition, a degree of personalization in communication can be defined according to the above parameters, so that the alternatives outlined in the definition of the contextual VIUE model are proposed.

Finally, it is worth noting that the proposed model is totally dynamic, i.e., the weights of each of the criteria can be modified dynamically, so that, for example, depending on the month or even during the day, and for the B2B model we are referring to. The needs of attention to incidents in times, such as tax filing, are much higher than at other times of the year, or, due to the schedule, the incidents usually have peaks and valleys during the day; the priorities can be adapted dynamically.

That is why for this case, we believe that the sensitivity analysis, Section 3.2.5, can be omitted as we are prioritizing interactions according to a series of attributes and there are two determining variables, the waiting time, which measures response tolerance in relation to the type of customer, and the workload of the Contact Centre.

Furthermore, we can extend the model to any other type of business in both B2C and B2B modes, in which case, perhaps the criteria defined can or should be extended, as well as adjusting the priorities defined for each criterion according to the Contact Center's needs.

A possible improvement of the current model could be to obtain customer value through the Customer Engagement Value (CEV), based on their purchasing (CLV), influence capacity (CIV), recommendation capacity (CRV), knowledge generation capacity (CKV) and the service they provide (CSV).

## 7. Conclusions

In the literature, the models developed for Contact Centers are based on the prioritization of calls according to variables related to the ITIL model [11,20] (Impact and Urgency) and the management of waiting queues. In this paper, we have incorporated a model that improves existing models, prioritizing interactions whose attention is based not only on impact, urgency and waiting time management, but also incorporates values such as the emotional nature of the interaction [49], and the value of the customer [6]. Our literature review did not identify multi-criteria decision-making models in a linguistic domain of information representation, as is done in this paper.

The practical implications of the study are based on the development of a working methodology that allows the integration in a CRM/Contact Center tool for real-time data collection to prioritize interactions according to criteria such as customer value, importance, urgency, emotional nature of the interaction, as well as including the response to customer SLAs (waiting time) and attending to the workload of the Contact Centre. On the other hand, as found in the literature review, there is no methodology based on fuzzy logic and decision-making theory that provides an answer to the problem of prioritizing customer-brand interactions.

The metrics indicated above are proposed, although, in each case, the decision-maker(s) may choose to prioritize or develop an extended model, considering the criteria that best fit the typology of the business. The use case is applied to a B2B model of a management software company, where the criteria and their weights have been defined in common agreement between the different areas involved in the decision-making process.

By automating the Contact Center with tools that allow the integration of all the company's information (single data) and capable of providing real-time self-help to the agent and the customer for decision making, the company will undoubtedly improve in areas such as [3,50,51]:

- Reduce Contact Center TMO.
- Increasing customer perception, NPS.
- Automate repetitive agent actions through robotic process automation (RPA), use bots oriented to support the agent in their search and analysis efforts with the goal of better connecting emotionally with customers.
- Integrate collaborative workspaces, eliminating information silos, where experts can cooperatively solve problems.
- Apply AI (predictive) models to analyze and direct the customer to fast, real-time solutions.
- Reduce learning time for contact center agents by providing them with tools that enable them to obtain real-time information from the systems.
- Reduction of the abandonment rate, by reducing the TMO we relieve the agents of their workload.
- Increase in first call incident resolution (FCRR).

As can be seen, the literature related to ticket prioritization in the Contact Center is limited to aspects related to the following categories of studies, Table 2:

- Queue management.
- Routing.
- Service level.
- Training.
- Personalization.

There are no studies related to the use of a methodology based on multi-criteria decision-making oriented to the process of prioritizing incidents in the Contact Centre.

That is why the methodology used in this article is new in this sector, and on the other hand, it can be extended to any decision-making process in this and other business environments.

## 8. Future Works

In the work presented in this article, we have developed a model of prioritization and personalization of interactions with the Contact Center (VIUE), built upon the RFID model for assessing customer value from the perspective of the Contact Center.

Among the points for improvement and future lines of research resulting from the work carried out, the state of the art, knowledge of the sector and customer needs, the following can be highlighted:

- The measurement of customer value can be considered an aggregate of several factors: $CEV = f(CLV, CKV, CIV, CRV)$ [6]. It is advisable to include one more parameter, the customer value from the contact center point of view, RFID, is what we call Customer Service Value (CSV), so that CEV is expanded with CSV,

$CEV = f(CLV, CKV, CIV, CRV, CSV)$. An aggregated and weighted measurement process would strengthen the CEV model.

- The employee attrition rate is a metric that contact centers are concerned about due to the high turnover in the industry, which is often attributed to the demanding work and emotional requirements [52]. Using a procedure that allows, predicting and interpreting the abandonment rate of contact center personnel would be a very important challenge.
- Apply RFID and VIUE models to different business environments, focusing on retail, insurance, banking, services, healthcare, and tourism. Each applied case will contribute to strengthen and possibly expand each of the models with specific characteristics of each sector.
- Create a communication add-on based on the VIUE model, between the Contact Center platform and the CRM, to define the interaction prioritization parameters in a totally dynamic way.
- Use Artificial Intelligence (AI) in the recommendation process so that, based on the prioritization of the interaction and the customer's value, personalized recommendations can be established.
- Extension of the current study incorporating multi-expert decision-making, applying the fuzzy AHP model (FAHP).

**Author Contributions:** Conceptualization, G.M.D.; data curation, G.M.D.; formal analysis, G.M.D.; investigation, G.M.D.; methodology, G.M.D. and R.A.C.G.; project administration, G.M.D.; resources, G.M.D.; software, G.M.D.; supervision, R.A.C.G.; validation, G.M.D.; visualization, G.M.D.; writing—original draft, G.M.D.; writing—review and editing, G.M.D. All authors have read and agreed to the published version of the manuscript.

**Funding:** This research received no external funding.

**Data Availability Statement:** The data presented in this study are available upon request from the corresponding author. The data are not publicly available due it is real data from a technology company.

**Conflicts of Interest:** The authors declare no conflict of interest.

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
