# Peer review of "Fuzzy Logic and Decision Making Applied to Customer Service Optimization"

_axioms, doi:10.3390/axioms12050448_

Round 1

Reviewer 1 Report

The authors proposed an integrated of fuzzy Multi-Criteria Decision-Making to customer services optimization. However, there are some issues that should be revised by the author.
- After an evaluation of this paper in ithenticate, I found a somewhat high rate of palgiarism from a paper already published by the same authors called "RFID: A Fuzzy Linguistic Model to Manage Customers from the Perspective of Their Interactions with the Contact Center". Authors should minimize the rate of plagiarism from this paper.
- The authors should be explained more the main contributions and research gap of the paper, in particular, in the introduction section, more than the present explanations.
- The relevant and recent references should be added in the paper.
- The research gap is unclear. A comprehensive table should be presented by the authors to show the literature review based on their assumptions, methods, and results.
- fuzzy logic is based on the use of triangular or trapezoidal fuzzy numbers, in the paper the authors used fuzzy AHP but in the presentation of this method they presented the classic AHP. So I propose to adapt AHP to fuzzy AHP (rating scale, equations...).
- I propose to the authors to make in the second section a sub-section for "Multi-criteria decision making applications in customer services".
- How we can judge about these results? Comparisons with existing models from the literature are missing. Discuss your improvements.
- A sensitivity analysis is missing.
- In the last section, the authors should elaborate more on the practical implications of their study, as well as the limitations of the study.

Author Response

Thank you for conducting this review and for each of your enriching comments.

Best regards.

Reviewer 2 Report

I found this quite an interesting manuscript to read. The application area is clearly one of commercial interest at an international level.

The Introduction nicely explained the idea behind the work, while the Literature Review contextualised the research.

I did find the Methodology section a little bit of struggle to read with some complicated theory/definitions.

The references are wide ranging and appropriate .

The results presented in particular in Tables 6 to 11 were a little confusing to me, especially with respect to 'u'.

Rather than a long Discussion, there should be a Discussion section and a Conclusion section.

The authors should address a slight issue whereby frequently double words appear in particular "Figure Figure" and "Table Table".

Author Response

(The authors gave the same response as above.)

Reviewer 3 Report

The manuscript proposes to apply a model of unification of heterogeneous information in 2-tuple linguistic evaluations, to obtain the global evaluation of each interaction by applying the Analytic Hierarchy Process (AHP). The model has a dynamic process of prioritization of interactions. The paper is generally well-written, and the contributions are clearly stated. However, there are some grammatical and format errors the authors need to address. The authors should carefully proofread the manuscript. Some of the problems are as follows:

1.      Line 25, “A data”, the word data is the plural form of datum.

2.      The values of CR in the manuscript are using , instead of “.”.

3.      The captions of figures have inconsistent fonts.

4.      The image quality of Figure 7 should be improved.

Author Response

(The authors gave the same response as above.)

Reviewer 4 Report

The authors propose to apply a model of unification of heterogeneous information in 2-tuple linguistic evaluations, to obtain the global evaluation of each interaction by applying the Analytic Hierarchy Process, and in this way be able to have a dynamic process of prioritization of interactions.

Since the concept of the membership function is fundamental in fuzzy logic, the authors could better detail the use of these functions in their proposal. Membership functions were mentioned only in the section 3.3.3.

In Discussion section, the authors should highlight their contributions, comparing the results with those found in the literature.

Author Response

(The authors gave the same response as above.)

Round 2

Reviewer 1 Report

Thank you, I suggest to accept this paper